# Developments in the Surgical Approach to Staging and Resection of Rhabdomyosarcoma

**DOI:** 10.3390/cancers15020449

**Published:** 2023-01-10

**Authors:** Sheila Terwisscha van Scheltinga, Timothy Rogers, Naima Smeulders, Federica deCorti, Florent Guerin, Ross Craigie, Gabriela Guillén Burrieza, Ludi Smeele, Marinka Hol, Rick van Rijn, Joerg Fuchs, Guido Seitz, Andreas Schmidt, Beate Timmermann, Per-Ulf Tunn, Cyrus Chargari, Raquel Dávila Fajardo, Olga Slater, Jenny Gains, Hans Merks

**Affiliations:** 1Princess Maxima Center for Pediatric Oncology, 3584 CS Utrecht, The Netherlands; 2Department of Pediatric Surgery, Bristol Royal Hospital for Children, Bristol BS1 3NU, UK; 3Great Ormond Street Hospital for Children NHS Foundation Trust, London WC1N 3JH, UK; 4Pediatric Surgery, Women and Children’s Health Department, University Hospital of Padua, 35128 Padua, Italy; 5Department of Pediatric Surgery, Bicetre Hospital, Paris-Saclay University, Assistance Public Hospital of Paris (AP-HP), 94270 Le Kremlin Bicetre, France; 6Paediatric Surgery, Royal Manchester Children’s Hospital, Manchester University NHS Foundation Trust, Manchester M13 9WL, UK; 7Department of Pediatric Surgery, University Hospital Vall d’Hebron, 08035 Barcelona, Spain; 8ENT Department, University Medical Center Utrecht, 3584 CX Utrecht, The Netherlands; 9Department of Radiology and Nuclear Medicine, Amsterdam UMC, University of Amsterdam, 1105 AZ Amsterdam, The Netherlands; 10Department of Pediatric Surgery and Pediatric Urology, University Children’s Hospital Tuebingen, 72076 Tuebingen, Germany; 11Department of Pediatric Surgery and Urology, University Hospital Marburg, Baldingerstraße, 35043 Marburg, Germany; 12West German Proton Center, University of Essen, 45147 Essen, Germany; 13Tumour Orthopedics, HELIOS Klinikum Berlin-Buch, 13125 Berlin, Germany; 14Department of Radiation Oncology, Gustave Roussy Comprehensive Cancer Center, 94805 Villejuif, France; 15Department of radiation oncology, University Medical Center Utrecht, 3584 CS Utrecht, The Netherlands; 16Department of Oncology, University College London Hospitals NHS Foundation Trust, London NW1 2PG, UK; 17Division of Imaging and Oncology, University Medical Center Utrecht, 3584 CX Utrecht, The Netherlands

**Keywords:** surgery, rhabdomyosarcoma, developments, staging, local treatment

## Abstract

**Simple Summary:**

Although survival after rhabdosarcoma treatment has improved over the years, one third of patients still develop locoregional relapse. This review aims to highlight developments pertaining to staging and local treatment of specific RMS tumor sites, including head and neck, chest/trunk, bladder-prostate, female genito-urinary, perianal, and extremity sites. It also aims to address the local treatment strategies of subgroups of patients who are very young, or those with metastatic or relapsed disease. Specific surgical techniques like whole limb perfusion, HIPEC and fluorescence guided surgery are discussed separately. The goal of the innovations is improving loco-regional control of disease, whilst minimizing morbidity of treatment strategies.

**Abstract:**

Although survival after rhabdosarcoma treatment has improved over the years, one third of patients still develop locoregional relapse. This review aims to highlight developments pertaining to staging and local treatment of specific RMS tumor sites, including head and neck, chest/trunk, bladder-prostate, female genito-urinary, perianal, and extremity sites.

## 1. Introduction

Rhabdomyosarcoma (RMS) outcomes depend on accurate diagnosis, staging, and risk-adapted multimodal local and systemic therapy. Failure of treatment most commonly occurs secondary to locoregional disease relapse; hence effective therapy of the primary tumor, as well as accurate identification and treatment of pathological lymph nodes, are of paramount importance. Of approximately 85% of patients present with non-metastatic disease, up to one third will relapse, of whom two thirds do so at the primary site or in regional lymph nodes [1]. In patients presenting with metastatic RMS, two thirds will progress or relapse, with most progressions/relapses occurring in several locations, including the primary site in over half. With few exceptions, relapse or progression carries a poor prognosis. Hence, the search for more effective treatment strategies for loco-regional disease to prevent relapse, as well as better systemic therapies for metastatic disease continues. In the rapidly expanding era of personalised medicine, matching biological agents to patient-specific tumor biology will bring about new, more targeted therapies. Tumour biology may also enhance patient specific rationalisation of the amount of treatment required for cure. This means that the diagnostic biopsy must provide sufficient tumor tissue for a comprehensive bio-molecular analysis. 

This review aims to highlight the developments in staging and local treatment for specific RMS tumor sites, including head and neck, chest/trunk, bladder-prostate, female genito-urinary, perianal, and extremity sites. For the latter, Isolated Limb Perfusion (ILP) will be described. The use of Near-InfraRed Spectroscopy with IndoCyanine Green (NIRS/ICG) to better visualise the primary tumor for targeted surgery will be explored. The role of sentinel lymph node biopsy and the increasing use of NIRS/ICG to identify pathological lymph nodes will be discussed. 

The important interdependence of systemic therapy, surgery, and radiotherapy will be highlighted by drawing attention to the timing and sequence of these combined treatment modalities. 

Finally, the review will seek to address the local treatment strategies for particular subgroups of patients, such as the very young, and those with metastatic or relapsed disease. For infants, the experience with HIPEC (hyperthermic intraperitoneal chemotherapy) will be evaluated. Amongst the patients with metastatic or relapsed disease there will be some who benefit from surgical resection; recognising to whom to offer salvage surgery is important so as to avoid major surgeries in those where this is futile. 

## 2. Diagnosis and Staging

Magnetic resonance imaging (MRI), including diffusion weighted imaging (DWI), is the principal imaging modality for RMS and should be performed according to international guidelines [2]. This allows the collection of homogeneous datasets that can be used for assessment of tumor response after induction therapy. At diagnosis, the aim of MRI imaging is to delineate and characterise the tumor, describe the relationship of the tumor with its surrounding structures, and identify the presence of regional and distant tumor spread. 

Staging of RMS requires [F-18]2-fluoro-2-deoxyglucose (FDG) positron emission tomography (PET)/CT or PET/MRI (lymph node metastases and distant metastases) and chest CT (pulmonary metastases) in combination with surgical staging (confirmation of involvement), bone marrow aspirations and trephine biopsy [3,4]. Following induction therapy, previous studies have shown that volume response is not a valid biomarker for RMS outcome and therefore there is an urgent need to identify a valid surrogate biomarker of outcome to guide further therapy [5,6,7,8]. When present, metastatic disease should be measured in one dimension, according to RECIST 1.1 criteria [9]. 

Although imaging techniques are improving, invasive techniques are still necessary to confirm or refute radiological findings, such as for histological assessment of suspicious nodal involvement. Lymph node biopsy can upstage patients from N0 on imaging to N1 in 16–20% of patients, and downstage suspicious nodes to N0 in 25% of patients [10,11].

Lymph node excision or core needle biopsy (CNB) are preferred techniques to determine pathological lymph node status. The use of fine needle aspiration (FNAC) is not reliable, as shown in a meta-analysis comparing FNAC with CNB of axillary lymph nodes in patients with breast cancer [12]. Hence FNAC is not advised for lymph node assessment.

Sentinel lymph node biopsy (SLNB) is used to detect potential micrometastases in patients with radiologically N0 nodes. Typically, SLNB is performed for extremity tumors using a radioactive-tracer and blue dye, but can be used for other sites with a high propensity for nodal spread, such as head and neck non-parameningeal and perineal RMS. With the advent of surgical telescopes with near-infrared capabilities (NIRS), agents, such as indocyanine green (ICG), are gaining in popularity for SNLB, for instance in paratesticular RMS [13]. Likewise, advancements in imaging techniques, such as Tc-Tilmanocept, may improve the intraoperative detection of sentinel nodes when these are close to the primary tumor or for tumors with complex lymphatic drainage patterns [14].

## 3. Treatment of Tumors Arising at Specific Primary Sites

### 3.1. Head & Neck 

Local treatment is essential for control of head and neck rhabdomyosarcoma (HN-RMS). Biopsy confirms the diagnosis, using CNB or incisional technique. For most HN-RMS complete (R0), resection is not possible without compromising form and function and is therefore not recommended. Following induction chemotherapy, surgery can be performed in selected cases, if a complete resection (R0) is possible without organ impairment. In some cases, this precludes the need for radiotherapy (RT), however, the majority of patients with HN-RMS require RT [15]. 

In studies analyzing surgery in HN-RMS, non-parameningeal (NPM) tumors were more often resected compared to parameningeal (PM) tumors. The role of surgical resection is limited to selected cases and requires a multidisciplinary approach. Dumbrowki et al. showed a lower mortality rate when surgery could be added to radiotherapy and chemotherapy [16]. A Parisian group has shown improved survival with the addition of surgery in infratemporal and pterygopalatine fossa tumors [17]. 

To minimise late adverse events, a specialised local treatment method-AMORE (Ablative Surgery, Moulage brachytherapy and Reconstruction) was developed in the Netherlands in the 1990s [18]. The advantages of this technique are (1) that surgery is limited to a macroscopic resection preserving most organ functions, and (2) that brachytherapy (BT) delivers the most conformal dose to the tumor bed with rapid dose fall off beyond the target volume, thereby minimising the volume of healthy tissue being exposed to irradiation. AMORE has been shown to afford similar survival to other local treatment options and carry less late adverse effects compared to external beam radiotherapy (EBRT) using photons [19]. Careful selection of patients for this technique is required. 

Contemporaneous advanced EBRT techniques, such as intensity modulated radiotherapy (IMRT) or volumetric modulated arc therapy (VMAT), as well as different EBRT modalities, such as magnetic resonance image guided radiotherapy (MRgRT) or pencil beam scanning intensity modulated proton therapy (IMPT), can provide a dose reduction to the surrounding normal structures while respecting the target volume coverage, and therefore can be considered for PM and NPM [20,21]. 

Late adverse events are common and varied in HN-RMS survivors: nearly 80% suffer events of severity grade 2 or higher based on the Common Terminology Criteria for Adverse Events (CTCAE) grading system [22]. The most common late adverse effects are musculoskeletal deformities, eye sequelae such as cataract and eyelid dysfunction, hearing impairment, and speech impairment [23,24]. Furthermore, Quality of Life (QoL) is known to be reduced in survivors of HN-RMS, specifically reporting low appearance and functional scores [23,25]. Owing to the diversity and high prevalence of late adverse effects, late effects clinics must offer a wide variety of specialists to address specific adverse effects and coordinate care for both parents and survivors. To minimise late adverse effects and improve QoL, a patient-specific and tumor-specific treatment approach should be implemented.

### 3.2. Chest/Trunk

Chest and trunk RMS have a poor prognosis. The exception is the recently described spindle cell rhabdomyosarcoma with fusion gene NCOA2/VGLL2: usually congenital and non-metastatic, occurring most commonly in the chest wall and trunk, it does not require RT [26]. Performing molecular characterization before taking any therapeutic decision is strongly advised in this scenario. 

The majority of chest wall RMS tumors are resected following neoadjuvant chemotherapy. Multidisciplinary preoperative planning is essential and should utilise recent advances in imaging, 3D modelling and include plans for reconstructive surgery. The role of R0 versus R1 resection for thoracic tumors is controversial, however, there is evidence of improved survival with R0 resection [27,28]. As a result of advances in prosthetic materials and reconstructive methods, an extensive R0 should be the goal, as long as it does not cause major risks or sequelae [29]. 

An experienced multidisciplinary surgical team is essential. Most abdominal wall tumors can be completely resected, taking the full abdominal wall thickness en-bloc, including the skin [30,31]. Reconstruction, using tailored myocutaneous flaps with or without mesh, achieves excellent results. Biological materials can provide benefits over previous alternatives [32].

Newer radiotherapy modalities, such as brachytherapy (BT), IMRT/VMAT, MRgRT or IMPT, are possible when R0 surgery is not feasible [33,34]. Although experience is still limited, these seem to reduce the sequelae commonly related to irradiation to these locations, in particular scoliosis and restrictive pulmonary disease [35].

### 3.3. Liver/Biliary

Biliary rhabdomyosarcoma (BRMS) represents approximately 1% of all childhood RMS, usually with a botryoid or embryonal histotype. Five-year survival for these subtypes is around 65–85% overall survival (OS) and 60–76% event-free survival (EFS) [36,37]. In a Cooperative Weichteilsarkom Studiengruppe (CWS) study, all children with the botryoid subtype survived. Factors for a worse outcome were age >10 years, alveolar histology, and tumor size >5 cm [38].

Diagnostic workup requires MRI/MRCP (Magnetic resonance cholangiopancreatography) [35]. Biliary drainage is advocated in case of jaundice to allow safer administration of chemotherapy. Tumor biopsy is essential for the definitive diagnosis and can be performed during the biliary drainage procedure by endoscopic retrograde cholangio-pancreatography (ERCP) or by mini-forceps inserted through the biliary drain, if transhepatic drainage is easier. Transhepatic biopsies should be performed with simultaneous biliary drainage, to avoid bile leakage. When interventional radiology or endoscopy is not feasible, biopsies can be performed via laparoscopy/laparotomy. Sufficient material (fresh tissue) should be obtained for diagnosis, biology and research according to RMS protocols. 

Treatment consists of neoadjuvant/adjuvant chemotherapy and local treatment (EBRT and/or surgery). Neoadjuvant chemotherapy is the initial approach of choice with a high tumor response rate shown in most trials. The value of maintenance chemotherapy is unclear. Optimal local treatment continues to be debated with regards to performing surgery alone, EBRT alone, or surgery in combination with EBRT. Although no difference in OS or EFS survival was observed for the timing of surgery in recent European studies (European paediatric soft tissue sarcoma Study Group (EpSSG) RMS-2005 and CWS) as well as on meta-analysis, a clear advantage for a delayed primary resection over upfront surgery was demonstrated with a significantly higher percentage of complete tumor resections (60% vs. 23%) and significantly reduced complications [36,37,38]. The surgeries employed ranged widely from simple tumor excision with portoenterostomy, advanced surgical procedures such as ‘Whipple’ procedure with/without liver resection, to liver transplantation. EBRT is pertinent in preventing relapse in patients receiving chemotherapy alone and in cases of incomplete surgical resection [39], although it has not been possible to demonstrate conclusive benefit on outcome for EBRT alone, the combination of surgery plus EBRT or chemotherapy alone. However, providing the statistical power to answer such questions is challenging, particularly for those with incomplete resections [40]. The total dose of EBRT (36 to 54 Gy) depends on protocols and resection margins. In the COG trial, patients with R0 or R1 residual disease received only 36 Gy (compared to up to 41 Gy, EpSSG regimen). Advanced highly conformal EBRT techniques, and modalities, such as proton beam therapy (PBT), are an effective therapy option in BRMS cases failing chemotherapy or for unresectable tumors. However, clinical data on PBT in BRMS is still scarce [41]. Further prospective international trials will be required to answer the unresolved controversies, including whether RT can be omitted in those with a botryoid subtype following complete response to chemotherapy or after complete tumor resection. In all other scenarios, the combination of surgery and EBRT remains favoured. 

### 3.4. Bladder/Prostate/Female Genito-Urinary Organs 

The key surgical developments in staging and local treatment of both bladder/prostate RMS (BP-RMS) and female genitourinary RMS (female GU-RMS) can be summarised into the adoption of:

Combination treatments for functional organ preservation as the gold standard for local therapy, early consideration of fertility preservation and minimally invasive techniques for staging, fertility preservation and local therapy.

Biopsy of the primary tumor must yield sufficient tissue to enable molecular fusion analysis. Primary tumor resection is discouraged except for very small tumors, e.g., those limited to the bladder dome in BP-RMS or a single exophytic polyp in female GU-RMS.

Where lymph-node involvement (N1) is demonstrated on cross-sectional and FDG-PET imaging, surgical confirmation is not valuable. However, any equivocal nodes should be assessed [42]. Minimally invasive (retroperitonoscopic, transperitoneal laparoscopic or robotic assisted) nodal sampling enables rapid recovery for timely commencement of chemotherapy [43]. ICG fluorescence may help the surgeon to better characterize the suspected nodes.

The desire to preserve function, and not just organs, has resulted in a revolution in local therapy for BP-RMS. Martelli and Haie-Meder’s ground-breaking concept aimed to surgically debulk the tumor remaining after initial chemotherapy to an area/volume that could reliably be covered by brachytherapy (BT) [44]. In view of the rapid dose fall-off, placement of radioactive sources directly within or next to the tumor without need for additional margin, makes BT the most conformal type of radiation treatment with the least effect on normal neighbouring tissues [45]. Excellent oncological outcomes and much improved functional results have made conservative surgery and interstitial BT (CS-BT) the gold standard local therapy in BP-RMS for those suitable. Across Europe different combinations have evolved combining conservative surgical approaches with BT of the macroscopic (R2) or microscopic (R1) tumor residuum, delivered at pulse-dose rate or high-dose rate [46,47,48]. Voiding day-and-night continence is achieved in 62–83%, with the assistance of anticholinergic medication in a proportion [45,46,47,48,49]. Strictures are reported between 1–15% and erectile dysfunction between 0–10% [47,49,50,51].

For BP-RMS patients, who are not suitable for CS-BT, local treatment consists of either radical surgery and/or EBRT. In North America, RT has traditionally been the favoured modality for local control of BP-RMS, although a reduced dose of RT has recently been shown to maintain equivalent oncological outcomes when combined with delayed surgery in intermediate risk BP-RMS patients [52]. 

In female GU-RMS, local therapy can be omitted for girls without evidence of residual disease on MRI and vaginoscopy after 4 to 6 cycles of chemotherapy. In patients with an incomplete response to chemotherapy, local control is achieved by conservative tumor resection and/or RT [42]. BT has gained great prominence, especially for vaginal and cervical tumors [53]. Late toxicities following BT (e.g., vaginal stenosis, sterility) were reported in 67% of cases in a series of 2D BT where the initial extent of disease was treated. Urinary late effects occurred in 45% of patients, including urethral or ureteral stenosis and incontinence [54]. Today, BT relies on 3D imaging, stepping source technology for greater personalization and targets only the tumor residuum after chemotherapy. Radical surgery such as hysterectomy for a tumor in the corpus uterus remains rarely indicated [42]. If feasible, trachelectomy (i.e., partial colpectomy) aiming for R0 resection for localised RMS in the upper vaginal or cervix, may be performed rather than BT or EBRT [55]. BT, however, remains the preferred irradiation modality when conservative surgery is not feasible [55].

A further concern is the impact of RT or chemotherapy on gonadal function. Options to circumvent infertility include transposition of the ovaries and testes or cryopreservation of oocytes, sperm, ovarian or testicular tissue [56,57].

### 3.5. Perineal/Perianal

RMS of the perianal/perineum is an unfavourable site (half survive) and includes tumors arising below the pelvic floor not from genito-urinary structures [58]. They are rare (1–2% of RMS) and tend to affect males <10 years, or females ≥10 years with FOXO1 fusion-positive tumors [59,60].

Lymph node spread affects nearly half of patients, making accurate surgical staging mandatory [61]. Nodal relapse can occur, especially in the iliac nodes, so the use of laparoscopy and sentinel nodal staging techniques may allow more accurate treatment [62].

Even small tumors should be biopsied first because upfront resection is seldom microscopically complete. Induction chemotherapy is followed by individualized local therapy. Because loco-regional relapse is the commonest cause of treatment failure, combinations of EBRT and BT with surgery need development to both improve survival and maximize functional outcomes [62,63]. Intra-operative adjuncts, including endorectal ultrasound and electrical muscle stimulation, may differentiate a tumor from a sphincter to decrease surgical morbidity. Specialist follow ups with survivors should be offered to uncover continence problems that may respond to treatments such as pelvic physiotherapy and sacral nerve stimulation.

## 4. Local Therapy in the Very Young 

Challenges in local control are particularly pertinent to infants and very young children due to the disproportionate size of the tumor to the body mass and the often permanent, devastating side effects of EBRT. These difficulties, coupled with reduced doses of chemotherapy to avoid excessive toxicity to immature organs, has resulted in inferior outcomes in this age group [64,65,66].

Complete surgical resection (R0) has been reported as a strong prognostic factor for infants with RMS by the CWS group [67]. However, 33% of surviving patients suffered long term impairment because of amputation/cosmetic deficit or urinary tract malfunction.

Improved outcomes in infants have been reported, more recently, in the EpSSG 2005 study, where OS for localised RMS reached 88.4%, significantly higher than in the older age group. This puts further emphasis on ensuring avoidance of mutilating surgeries and restricting EBRT. In the 2005 EpSSG study, 33.6% of infants received RT, which is an increase compared to previous SIOP (International Society of Pediatric Oncology) studies, but in 41.7% this was delivered as BT [68]. Conservative surgery with BT is established therapy for selected patients with bladder/prostate and perianal RMS with results comparable to standard surgery and EBRT but with preservation of function (Figure 1) [33,48,50]. If EBRT in very young patients is deemed unavoidable, proton beam therapy might provide best sparing of organs at risk [69].

HIPEC (hyperthermic intraperitoneal chemotherapy) and cytoreductive surgery in young children was pioneered and reported by the Tubingen group, who successfully treated 6 patients aged 2–4.2 years with IRS III and IV group intraabdominal embryonal rhabdomyosarcoma. The team used reduced doses of cisplatin. At a median follow up of 12 months (range 7–41 months), all patients were disease free, with no severe (grade 3 or 4 CTCAE) side effects [70]. Additionally, a new animal model was developed to evaluate HIPEC in paediatric patients [71]. Several additional case reports in children aged 4 months onwards, who were successfully treated using HIPEC, support the need to investigate this approach in a systematic manner. 

## 5. Timing of Local Therapy

RT is a part of the multimodality treatment strategy and can be applied in combination with surgery. There are limited data comparing results between preoperative and postoperative RT in children. In adults preoperative RT can limit the target volume, allowing sparing of the surrounding tissue, and therefore potentially minimising long term side effects. However, it can lead to more serious wound complications [72]. This aspect is currently being studied in a randomized fashion within the ongoing EpSSG overarching study for children and adults with Frontline and Relapsed RhabdoMyoSarcoma (FaR-RMS) (NCT04625907). 

## 6. Fluorescence Guided Surgery

Inadequate surgical resection margins in RMS increases local recurrence and decreases survival, whereas unnecessary removal of normal tissue may increase treatment-related side effects and negatively impact function. Real-time intraoperative visualization of tumor using fluorescence with near-infrared light may aid the surgeon to accurately discriminate between healthy and malignant tissue. Fluorescence-guided surgery (FGS) has shown promise in hepatobiliary cancer and has been proposed for intraoperative delineation of other solid tumors. The differential of enhanced permeability and retention (EPR) between normal and tumor tissue allows demarcation of the tumor; whilst the newly formed more porous blood vessels in tumor tissues allow fluorescence molecules to accumulate, the poorly developed tumoral lymphatics result in increased fluorescence molecule retention. Persistence of fluorescence in tumor, whilst getting washed out of neighbouring tissues, enables the tumor lesions to be visualized to a tissue depth of 1 cm [73].

In future, the development of targeted fluorescence using tracers that bind to tumor-specific receptors may further enhance FGS. These tracers need to be tumor-specific and with a high percentage of the tumor cells expressing the target. So far, not many such receptors have been identified in RMS. However, of interest for RMS are the CD56, IGF-1R, and VEGF-A receptors. In contrast, B7-H3 and TEM1 have shown less promise for FGS in RMS [74]. 

Fluorescent agents have an emerging use in SLNB (Figure 1). A meta-analysis comparing ICG with blue dye showed a clinically significant difference in detection rate of sentinel nodes in favour of ICG, for breast, dermatological, and gynaecological cancer. In RMS the use of ICG as a tracer in sentinel node procedures shows promising results but requires further validation (Figure 2) [75].

## 7. Extremity Tumors and Isolated Limb Perfusion 

Although not part of routine care, isolated hyperthermic limb perfusion (ILP) with recombinant human tumor necrosis factor alpha (TNF-α) and melphalan is a treatment option to avoid mutilating surgeries and amputations for locally advanced RMS of the extremities that have shown a poor response to systemic neoadjuvant chemotherapy, in local recurrence, or in a palliative setting [76]. ILP serves as a local treatment and has no effect on overall survival or development of distant metastases. It achieves 15- to 25-fold higher regional concentrations of TNF-α and melphalan than can be by systemic administration, without systemic side effects [77]. The procedure is performed under mild hyperthermia (38.5–39.5 °C) for 90 min. TNF-α leads to increased vascular permeability of tumor vessels resulting in increased concentrations of cytotoxic drugs within the tumor and selective destruction of tumor associated vessels by apoptosis and inflammation [78]. ILP is followed by tumor resection 6–8 weeks later.

In a systematic review of 18 studies including 1030 patients (age ≥ 12 years), the limb salvage rate was 81%, the overall response rate 71%, and a complete tumor response observed in 22% of cases. The rate of amputation due to complications was 1.2% [79].

Further work is required to ascertain whether the use of ILP may allow decreased use of RT in young patients, thereby reducing the risk of radiation induced second malignancies. Given the limited experience of ILP in children and adolescents, this technique should only be performed in selected centers.

## 8. Role of Local Therapy for Metastatic Disease 

There is little evidence on the optimal treatment for metastatic sites of disease, which is the subject of ongoing research (FaR-RMS trial). Current standard of care recommends systematic irradiation of all metastatic sites that can feasibly be treated without incurring bone marrow failure.

Selection bias is likely to be a factor in the observed improved OS of patients receiving aggressive local treatment (both surgery and RT) to the primary tumor compared to patients treated exclusively by surgery or RT alone [80]. In some series, local treatment of all metastatic sites was associated with improved survival, and the use of whole lung radiotherapy improved pulmonary control in patients with lung metastasis [81,82,83]. In the BERNIE study metastatic RMS patients showed significantly improved survival when they received radiation therapy. As this was not a randomized analysis, it is unclear whether this was related to a positive effect from irradiation or a difference in the patient population who received radiation therapy. Radical treatment of the primary site significantly improved survival, confirming the importance of radical treatment of the primary site. Irradiation of all disease sites versus only some sites increased overall survival, suggesting that irradiation of metastatic sites may also be important. This is subject of a randomized study question in the ongoing FaR-RMS study (ClinicalTrials.gov Identifier: NCT04625907; accessed 17 September 2020) where patients with adverse risk profile will be randomized to yes or no receive radiotherapy to metastatic sites, where all will receive local therapy to loco-regional disease. In general, there is little evidence for a role for surgery in treating metastatic sites, although there may be a role in cases where surgical resection may reduce radiotherapy dose to vital organs at risk. Where there is no role for surgery in the evaluation of indeterminate pulmonary nodules at diagnosis [84], residual lung nodules after induction chemotherapy can be resected by thoracotomy or VATS (Video-assisted thoracoscopic surgery): peripherally located nodules can be localized, whilst for centrally located lesions, different marking techniques have been described (CT-guided marking with hook-wires/micro-coils, patent blue dye, lipiodol) [85,86,87,88,89]. Again, this is only indicated where surgical resection of remaining nodules leads to significant reduction of radiation dose to organs at risk. Recently the use of ICG has been investigated [90]. The risk of respiratory insufficiency should be considered for resections of extensive metastatic disease, and lung-function should be preserved [91].

In selected cases, an aggressive surgical approach even for metastases arising in rare sites, may improve survival [92]. As an alternative, when surgical resection is not feasible, ablative techniques (thermal, chemical, irreversible electroporation) are frequently used in adult patients. A few reports have shown radiologically guided ablative techniques to be feasible in children, highlighting the need to further define their role [93].

Isolated nodal metastases in RMS determine an intermediate prognosis between localised disease and systemic dissemination [94]. However, patients with FOXO1-fusion positive RMS with nodal disease have a prognosis similar to those with systemic dissemination [95]. According to protocol, lymph node metastases require RT, with surgery reserved for staging and when RT is contraindicated. 

## 9. Role of Surgery for Relapse Disease

The treatment of patients with recurrent RMS is challenging even though relapsed tumor is often smaller than disease at diagnosis. Biopsy of tumor relapse was not often performed but is now recommended in order to obtain material for molecular sequencing to guide potential targeted therapy [96]. Despite the prior use of surgery and/or RT in most first line treatment, repeat surgery and/or RT should be systematically considered [97].

Patients with local relapse who receive aggressive local treatment (surgery and radiotherapy) have better outcomes compared to patients who receive surgery only or no local treatment [98]. Treatment carrying higher risks or even mutilating procedures may be acceptable in recurrent disease compared to local treatment of patients at diagnosis [99,100].

Radiotherapy, too, can be used at recurrence, even in previously treated patients; salvage re-irradiation can reduce the risk of local progressive disease at the price of acceptable toxicity and risk of second tumors [101]. In HN-RMS recurrent disease, brachytherapy as part of the previous described AMORE-protocol has shown excellent outcomes [102].

## 10. Conclusions

Developments in the surgical approach to staging and resection in RMS continue to evolve. Intra-operative adjuncts to assist the surgeon in distinguishing a tumor or lymph nodes from surrounding tissue like fluorescence guided surgery promise to improve the quality of primary tumor resection, as well as to more clearly identify lymph nodes and distant metastases for biopsy or resection. The timing, sequence, and combination of conservative surgery and radiotherapy, and the use of new RT techniques, such as 3D BT and proton beam RT, optimises disease control and can minimise loss of organ function. These innovations promise to improve outcomes for patients by improving loco-regional control of disease, whilst minimizing morbidity of treatment strategies.

## Figures and Tables

**Figure 1 cancers-15-00449-f001:**
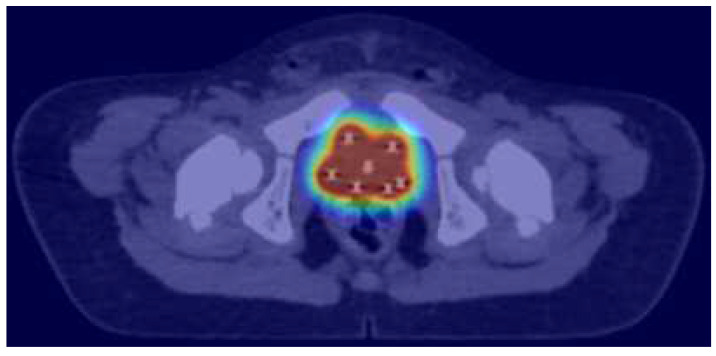
Brachytherapy dose distribution of intraoperative/trans-perineal brachytherapy implant for the treatment of bladder neck/prostate rhabdomyosarcoma: the red isodose surface corresponds to 100% of the prescribed dose, the yellow to the 80%, the green to the 50% and the dark blue to the 25%, respectively.

**Figure 2 cancers-15-00449-f002:**
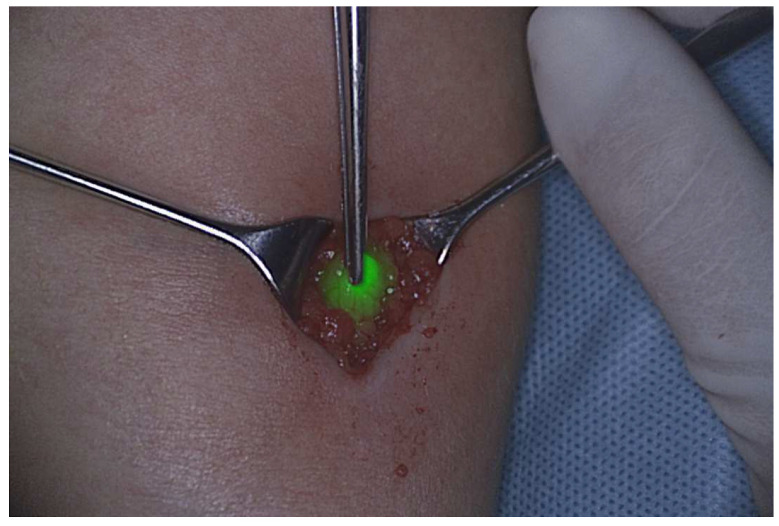
Sentinel node detected by near-infrared light.

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
