# Peer review of "Developments in the Surgical Approach to Staging and Resection of Rhabdomyosarcoma"

_cancers, 2023, doi:10.3390/cancers15020449_

Round 1

Reviewer 1 Report

This is a very comprehensive and mostly well written paper. I understand well that the focus is in the developments and all outcomes with new treatment modalities have not been thoroughly investigated and their benefits - if promising -not fully known.  I have the following points

1) BT (QD)  in female GU -RMS was earlier associated wit 67% rate of late toxicities, and 45% developed urinary incontinence or stenosis. However do we have any data that 3D BT version is any better in this ? 

2) In perineal perianal disease relapse is common especially in older children - is relying too much on new development ( BT,  endorectal ultrasound, muscle simulation,fluorescent guidance  etc) to preserve function by jeopardising the radicality of the primary surgery, BT may be detrimental to nala function and cause strictures

3) Are the authors able to estimate which of the new developments are most promising in improving or maintaining present survival rate with less mutilation and morbidity 

Author Response

Dear reviewer

thank you for your comments

1. 

We agree with the reviewer on the fact that BT can be associated with the development of late toxicity. It very much depends on de implant sort and dosimetry distribution. The current (often MRI) image-guided planning resources facilitates a dosimetry optimization that may reduce the problem. A so called “good implant” will allow to reduce the chance of toxicity but it will never fully prevent the risk when the prescribed dose exceeds certain limit, as it is required for RMS. Within the ongoing FaR-RMS study, a BT quality assurance process is available which will permit the retrospective analysis of the 3D dosimetry parameters of prospectively collected data.

Regarding female GU RMS specifically an important aspect is the post-BT management of the patients. There is extensive research in the adult population with a gynecology malignancy treated with BT, either standing alone or in combination with EBRT, where different measures are applied in order to reduce the chance of vaginal adhesions/obliteration, for instance the periodical vaginal dilation during the first 6-12 months after completion of the treatment. However these measures are rarely applied to the pediatric population due to the burden that frequent physical examination creates. Aligning the recommendations to the adult population may help Improving the quality of life of the pediatric population. 

2. This could read,' Patients with unresectable disease require definitive RT. Others may undergo surgery, but unfortunately attempted complete resection is seldom achieved. Therefore, combinations of RT and surgery need development. Because  loco-regional relapse is the commonest cause of treatment failure, innovative combinations of EBRT and BT with surgery, need development to both improve survival and maximize functional outcomes.'

So we refrased the sentence 294 in this section:

Because loco-regional relapse is the commonest cause of treatment failure, combinations of EBRT and BT with surgery need development to both improve survival and maximize functional outcomes.

3. Some new techniques are further developed than others so easier to implement. in the conclusion section we added fluorescence guided surgery'' and 'new RT techniques like 3D BT and proton beam RT in sentence 455 and 458:

Intra-operative adjuncts to assist the surgeon in distinguishing tumor or lymph nodes from surrounding tissue like fluorescense guided surgery promise to improve the quality of primary tumor resection, as well as to more clearly identify lymph nodes and distant metastases for biopsy or resection. The timing, sequence, and combination of conservative surgery and radiotherapy, and the use of new RT techniques like 3D BT and proton beam RT optimises disease control and can minimise loss of organ function

Reviewer 2 Report

1. Nice summary of the role for surgery in RMS in multiple sites/locations.

2. Importance of role of combined modality with surgery, radiation, and chemo are highlighted appropriately

3. Decisions on surgery for metastatic disease are balanced.

4. Fluorescence guided surgery is interesting and as noted is not available or routinely done. 

Author Response

We would like to thank the reviewer for the nice comments below.

Kind regards

Sheila Terwisscha and Timothy Rogers